# Medication Regimen Complexity Index Score at Admission as a Predictor of Inpatient Outcomes: A Machine Learning Approach

**DOI:** 10.3390/ijerph20043760

**Published:** 2023-02-20

**Authors:** Yves Paul Vincent Mbous, Todd Brothers, Mohammad A. Al-Mamun

**Affiliations:** 1Department of Pharmaceutical Systems and Policy, School of Pharmacy, West Virginia University, Morgantown, WV 26506, USA; 2Department of Pharmacy Practice, College of Pharmacy, University of Rhode Island, Kingston, RI 02881, USA; 3Roger Williams Medical Center, Providence, RI 02908, USA

**Keywords:** medication regimen complexity, machine learning, critical care, clinical outcome, racial disparity

## Abstract

Background: In the intensive care unit, traditional scoring systems use illness severity and/or organ failure to determine prognosis, and this usually rests on the patient’s condition at admission. In spite of the importance of medication reconciliation, the usefulness of home medication histories as predictors of clinical outcomes remains unexplored. Methods: A retrospective cohort study was conducted using the medical records of 322 intensive care unit (ICU) patients. The predictors of interest included the medication regimen complexity index (MRCI) at admission, the Acute Physiology and Chronic Health Evaluation (APACHE) II, the Sequential Organ Failure Assessment (SOFA) score, or a combination thereof. Outcomes included mortality, length of stay, and the need for mechanical ventilation. Machine learning algorithms were used for outcome classification after correcting for class imbalances in the general population and across the racial continuum. Results: The home medication model could predict all clinical outcomes accurately 70% of the time. Among Whites, it improved to 80%, whereas among non-Whites it remained at 70%. The addition of SOFA and APACHE II yielded the best models among non-Whites and Whites, respectively. SHapley Additive exPlanations (SHAP) values showed that low MRCI scores were associated with reduced mortality and LOS, yet an increased need for mechanical ventilation. Conclusion: Home medication histories represent a viable addition to traditional predictors of health outcomes.

## 1. Introduction

Obtaining an initial medication history at admission can significantly influence clinical outcomes [1], yet traditional scoring systems, such as APACHE or SOFA, remain the preferred prognostic tools [2,3,4]. Indeed, medication reconciliation informs on the negative consequences of medication list discrepancies [5]. Complete medication histories can help prevent medication errors and adverse drug events (ADEs) [6,7], which are more common among intensive care unit (ICU) patients than hospitalized patients [8].

Estimates of the clinical significance of medical histories in terms of medication errors range between 11% and 59% [9]. Medication errors are the third leading cause of hospital readmissions in the United States [10] and a significant cause of morbidity and mortality [11]. One hundred thousand people die each year as a result of medication errors in hospitals and clinics [12]. The incidence of adverse drug events is 6.5 per 100 admissions, with 28% of them judged preventable [13]. About 100,000 people die each year as a result of medication errors in hospitals and clinics. Moreover, medication errors cost approximately USD 20 billion each year [12]. The stratification of these costs per race remains unknown to the best of our knowledge; a curious fact in light of extant racial health disparities in the US. As minorities incur more medication errors and adverse drug events than Whites in the US [14,15,16], it stands to reason that they should also bear higher costs.

To reduce the burden of medication errors and ADEs and their costs, the assessment of accurate medication histories is essential. For this purpose, pharmacists capture medication regimen complexity, which encompasses medication dosage, frequency, and route [17]. In clinical pharmacy practice, the most commonly used tool for the assessment of medication regimen complexity is the Medication Regimen Complexity Index (MRCI) [18]. Although initially developed and validated in outpatient settings [19,20], the MRCI has been validated across various other medical settings and populations, thus making it a gold standard for assessing medication regimen complexity [18]. Previous findings show that MRCI is correlated with health outcomes, specifically prognosis (of diabetes and hepatocellular carcinoma) but also mortality [21]. Two different studies utilized registry data and hospital records to show that high MRCI scores were associated with higher 4-year mortality, and higher odds of mortality (adjusted odds ratio = 1.12) [22,23], respectively. A recent study showed that higher MRCI was associated with increased mortality, a longer ICU length of stay (LOS), and the need for mechanical ventilation (MV) [17]. As the literature suggests that medication complexity scores are higher among Blacks compared to Whites [24], it is highly probable that the clinical burden exacted on minorities is also higher. This, however, remains to be explored.

When taken at admission, MRCI scores, which reflect patient medication histories, could help better predict outcomes of importance, especially among critically ill patients, who are subject to far more intense treatment. In clinical settings, traditional scoring systems, such as SOFA and APACHE II remain the primary indicators of disease severity [2]. In the ICU, the ratio of critical care pharmacists to patients is sub-optimal (the median number of day shift critical care pharmacists per ICU was one), and higher ratios are associated with unsafe patient care [25]. The potential usefulness of MRCI at admission for a critical care pharmacist is thus threefold: (1) to serve as a guide to reduce drug errors and adverse events by facilitating reconciliation; (2) to provide an indication of the potential outcome of hospital stay; and (3) to help mitigate the effects of the ICU pharmacist-to-patient ratio in a cost-effective manner. In the present study, we hypothesized that home MRCI can effectively predict ICU outcomes at admission. We developed machine learning (ML) algorithms to assess this hypothesis across three health outcomes (ICU mortality, LOS, and the need for MV) among critically ill patients. We also evaluated the racial disparities that stem from using MRCI for the foregoing clinical outcomes. Finally, we tested MRCI in combination with either APACHE II or SOFA as joint predictors for these outcomes, given the possibility of a synergistic predictive effect stemming from medication histories, physiological states, and organ failure assessments.

## 2. Materials and Methods

### 2.1. Study Design

This was a single-center, STROBE-compliant retrospective cohort study of 322 patients enrolled in the ICU of a 220-bed community hospital in Providence, Rhode Island, USA, between 1 February 2020 and 30 August 2020. Due to the retrospective nature of the data, informed consent was not deemed necessary, as all patient data were de-identified prior to use. The study was a granted exemption by the Human Research Review Committee Roger Williams Medical Center (RWMC) Institutional Review Board (IRB: 00000058) and the University of Rhode Island Institutional Review Board (IRB: 00000559). The data were curated and reviewed for accuracy by the RWMC data-extraction team.

### 2.2. Data Sources

Patient-level data were extracted from the electronic medical record system of a medical ICU. The total of 322 patients aged at least 18 years included demographics, comorbidity scores, outcomes, medication counts, and individual medication components of the MRCI tool. Patients were excluded if they had an ICU LOS of less than 24 h, active transfer, or change in code status to hospice at 24 h.

### 2.3. Measures

#### 2.3.1. MRCI Score: Key Independent Variable

As medication regimen complexity is associated with several health outcomes, various tools have been developed to assess medication regimen complexity (Medication Complexity Index—MCI, the patient-level Medication Complexity Index—pMRCI, or the modified Medication Regimen Complexity Index—mMRCI) [26]. The MRCI is deemed the most reliable and valid of the currently available selection, in light of its good interrater and test–retest reliability [20,27]. The MRCI was calculated at the time of admission. The MRCI is a 65-item instrument divided into the following three sections: section A (32 items) for ascertaining dosage form; section B (23 items) for assessing dosing frequency, and section C (10 items) for evaluating additional directions (Appendix A) [28]. Each prescription drug, over-the-counter drug (orally or not orally taken) was weighted according to these three components [28]. The total score is the sum of the individual sections’ scores, and the higher the score, the more complex the medication regimen [26]. MRCI scores were manually calculated by an independent coder, and validated by Todd Brothers, using a random sample of the extracted data (20%).

#### 2.3.2. Outcomes

Three outcomes were evaluated: ICU mortality, LOS, and the need for MV. LOS was stratified into a categorical variable of two levels: those with LOS < 3 days and ≥3 days. The threshold of three days was taken from the mean average length of patients in the literature (~3.4 days) [29]. The need for MV (MV) was classified into two categories, namely, those that used MV and those that did not. Missing data were excluded (*N* = 3).

#### 2.3.3. Covariates

Variables of interest included age, Charlson comorbidity index score, gender, race, BMI, health insurance (public and private insurance), MRCI scores at admission, SOFA scores at admission, and APACHE II scores. With the exception of age, APACHE II scores, MRCI scores, and BMI, the remaining aforementioned independent variables were categorical. Race was stratified into Whites and non-Whites. Non-Whites or minorities encompassed Blacks, Asians, Hispanics, and unknown races; they were categorized as a single group because of their low sample size. Gender was recorded as “1” or “0”, for males and females, respectively.

#### 2.3.4. Machine Learning Models

Five machine learning models were designed, and each dependent variable was fitted to all models. The home medication model included age, gender, Charlson score, BMI, race, health insurance, and MRCI at admission. The MRCI was calculated using the information on the medications the patient was taking at home. The admission model substituted MRCI for APACHE II. The MRCI/APACHE model included APACHE II and MRCI as well as the demographics of the previous models. The SOFA model included SOFA and demographics. The MRCI/SOFA model included SOFA, MRCI, and demographics. Each model was evaluated across all cohorts and across the defined racial continuum (White and non-Whites). A detailed description of each model is available in Appendix A.

### 2.4. Model Development and Statistical Analysis

For descriptive statistics, the threshold of MRCI (low to high) was set at the third quartile value of MRCI scores. Classification learning algorithms build classifiers from a set of training data, and their performance is assessed based on how well they can predict unseen test data [30]. In the present work, four supervised classifiers were developed and tested for each model: logistic classifier (LC) [31], naïve Bayes (NB) [32], random forest (RF) [33], and extreme gradient boosting (XGB) [34]. The LC is a discriminative model that uses maximum likelihood parameter estimation, wherein a probability distribution is assumed about the data. The NB model uses the Bayes’ rule, which is a probability statement and simplifies the probabilities of the predictor values by assuming that all the predictors are independent of one another [32]. RF, a tree-based model, is an extension of the bagging method, which adds randomness to create an uncorrelated forest of decision trees with a random subset of features, thereby essentially ensuring low correlation among trees [33]. XGB is an implementation of Friedman’s stochastic gradient boosting algorithm, which encompasses classification and regression. Herein, weak classifiers are combined to produce an ensemble classifier with a superior generalized misclassification rate, while minimizing the loss function over numerous iterations [2,35,36].

The relative frequencies of classes may have a significant impact on the effectiveness of different models. In terms of our outcomes, there was an imbalance with regards to the proportion of dead/alive patients (alive: 76.1%, dead: 23%) and also between the racial groups (Whites: 64%, non-Whites: 35.1%). Practical approaches used to counter the influence of class imbalance on model output include model tuning, alternate cut-offs, and sampling methods, among other things [35]. In this study, we used sampling methods and model tuning to readjust sample sizes and improve model performances, respectively. The sampling method we used is termed the Synthetic Minority Oversampling Technique (SMOTE). SMOTE is an over-sampling approach in which the minority class is over-sampled by creating synthetic examples rather than by over-sampling with replacement [37]. Over-sampling of the minority class is performed by introducing synthetic examples along the line segments joining any of the k minority class nearest neighbors [37]. In this study, we used the five nearest neighbors. SMOTE was applied at a percentage of 400% to generate synthetic data with 1276 data points (from the original 319 instances) consisting of 786 Whites and 490 non-Whites. SMOTE interpolation was conducted prior to splitting the data into train and test sets. SMOTE was also applied after subsuming race into two major groups (White and non-White). We did not interpolate the original synthetic data based on race because, based on the initial sample size, we could not safely conclude that these were representative enough to create a synthetic dataset of such diversity.

Measures of association (correlation) were assessed using the Goodman–Kruskal test to account for factor and categorical variables [38]. Significant associations between age and BMI (r = −0.180), age and Charlson score (r = 0.341), and pre-admission MRCI and Charlson score (r = −0.279) were observed. However, as these were weak, no measures were taken to curb their potential yet unlikely influence on results. The data was partitioned into training (80% of the data) and test (20% of the data) sets.

To determine optimal classification results, a k-fold cross-validation for 100 repetitions was estimated. An automatic grid search algorithm was used to find the tuning parameters for the best ML algorithm [39]. For RF, this entailed selecting the mtry, the minimum node size, and the sample fraction. In XGB, variations in randomly selected hyperparameters (max_depth, eta, subsample, colsample_by_tree, and min_child_weight) allowed us to create 10,000 models to choose from. After using the XGB algorithm, we plotted the SHapley Additive exPlanations (SHAP) summary to help us interpret the findings. The SHAP or Shapley values method is a feature attribution technique that assigns to each feature a particular value for a particular prediction to assist with interpretability. As the Shapley value method utilizes features in every possible order to arrive at values, this allows for an unbiased interpretation of predictions [40]. The SHAP values were plotted for the entire sample and not across the racial divide.

Selected metrics for overall performance included the area under the receiver operator characteristic curve (AUC), the sensitivity (Se), the specificity (Sp), the positive predictive value (PPV), and the negative predictive value (NPV). Ninety-five percent confidence intervals (CI) were provided alongside these metrics. The area under the receiver operating characteristic (ROC) curve, or AUC, measures the predictive ability of learning algorithms’ ROCs. Sensitivity refers to true positives that are correctly predicted by the model, while specificity refers to true negatives that are correctly predicted by the model. Results are shown for the general population and across the racial divide. Only the best ML algorithm results are shown; the remainder is provided in the Appendix A. All analyses were performed using R (R Core Team, 2019). R packages Caret (for LC), e1071 (for NB), Ranger (for RF), and XGBoost (for XGB) were used for ML algorithms. Multicollinearity was assessed mathematically using package performance and graphically using the package ggally. SMOTE implementation used the package smotefamily, whereas graphs were produced using ggplot2.

## 3. Results

### 3.1. Descriptive Analysis

In total, data from 319 patients were included in the analysis (Table 1). The median age was 62 years and similar between low and high MRCI groups. There was no significant difference between the low and high MRCIs for BMI. No significant differences were detected between MRCI groups for APACHE II, the Glasgow Coma Scale (GCS), or the Simplified Acute Physiology Score (SAPS) II. The top conditions at admission included hypertension, acute renal failure, myocardial infarction, and metabolic encephalopathy (Table 1). The high MRCI group had the highest proportions of patients with these conditions, although the difference with the low MRCI group was not significant (Appendix A). The proportion of patients who died in the ICU was higher among the high-MRCI compared to the low-MRCI patients (26% vs. 22%). Appendix A shows that those with a high MRCI took more medications at home than those with low medications. Among those with high MRCI who survived, the distribution was broader compared to those who died. In Table 1, those with a low MRCI had a longer length of stay (113.2 h vs. 106.5 h) and needed more MV (65 h vs. 47.9 h) than those in the high MRCI group. This difference was not significant as emphasized (Appendix A). A heatmap (Appendix A) of the various medications taken at home by all patients shows that diuretics, genitourinary, and paralytic agents were highly used by those with low MRCI. Among those with high MRCI, IV fluids appeared to be slightly more predominant. Non-Whites had higher representation among those with high MRCI (31.9% vs. 21.4%) than Whites (Appendix A).

### 3.2. Model Performance: General Population

#### 3.2.1. ICU Mortality

Table 2 presents the performance measures for the best model using the best ML algorithm. The best ML algorithm throughout was XGB, and the best model was the MRCI/SOFA model. The performances across the remaining ML algorithms for all models can be found in Appendix A. For mortality, the predictive positive value (PPV) was defined as the percentage of patients predicted to die who in fact did die during their hospitalization. The home medication model showed a precision of ~60%, a prediction accuracy for patient death of ~70% and could accurately classify 73% of the time (Se) patients who had died. In this model, MRCI was the third-most important variable featured (Appendix A). Upon inclusion of SOFA, the MRCI/SOFA model outperformed all other models across all metrics with a precision of 100% and an overall prediction accuracy for patient death of 98%. Using this model, patients who died were correctly classified 96% (Se) of the time (Table 2 and Figure 1).

Despite the fact that SOFA drove the model and scored the highest on the variable importance chart, MRCI was the fourth most important variable, only outdone by age and BMI (Appendix A). The SHAP summary for the MRCI/SOFA model (Appendix A) showed that low scores of the MRCI were associated with remaining alive during an inpatient ICU stay.

#### 3.2.2. ICU Length of Stay

For LOS, the precision of the home medication model was 76%, whereas the prediction accuracy for patient LOS of less than 72 h was 68% (Table 2 and Figure 1). This model could accurately classify 64% of patients who were hospitalized for less than 72 h. As with mortality, the addition of SOFA (MRCI/SOFA model) provided the best results in terms of precision (95% CI: 74–83%), prediction accuracy (95% CI: 76–82%), and Se (95% CI: 79–83%). Variable importance featured MRCI in first and fourth place, across the home medication and the MRCI/SOFA models, respectively (Appendix A). SHAP values obtained for the MRCI/SOFA model showed that low values of MRCI were mostly consistent with a hospital LOS of less the 72 h (Appendix A).

#### 3.2.3. ICU Need for Mechanical Ventilation

Across all outcomes, the home medication model performed best for the need for the MV outcome. Accounting for a precision of 72%, a prediction accuracy of 75%, and a classification accuracy for patients that need MV of 78% (Table 2 and Figure 1), this model was not widely dissimilar in performance compared to the admission model or the MRCI/APACHE II model. Only the substitution/addition of SOFA in the SOFA and MRCI/SOFA models yielded far better outcomes. The MRCI/SOFA model had a precision of 84% (95% CI: 79–87%), a prediction ability of 87% (95% CI: 85–90%), and could accurately classify 90% (95% CI: 86–93%) of the time patients were in need for MV. In the home medication and the MRCI/SOFA models, MRCI was respectively the first and second-most important feature (Appendix A). Interestingly, the SHAP plot (Appendix A) showed that patients with low MRCI needed more MV.

### 3.3. Model Performance: Racial Continuum

Algorithm performance results for Whites are shown in Table 3, and for minorities in Table 4. Only the results for the best algorithms per model are presented in these tables; the remainder can be found in Appendix A. The home medication model performed worse for minorities than it did for Whites. For mortality, the home medication model performed better among Whites compared to the mixed population. It correctly classified patients who died 83% of the time, with a prediction accuracy of 81% and a precision of 78%. Among minorities, this model’s prediction accuracy was 73%. The home medication model was the second-best model among Whites after the MRCI&SOFA model, which yielded increased 95% CI for prediction accuracy (95% CI: 0.85–0.90), Se (95% CI: 0.82–0.93), Sp (95% CI: 0.86–0.93), PPV (95% CI: 0.87–0.93), and NPV (95% CI: 0.78–0.86).

For minorities, the SOFA and MRCI/SOFA models were the best in terms of Se and PPV, respectively. The best models among minorities were obtained using LC as opposed to XGB for Whites.

For LOS, the home medication model performed better among Whites with higher precision accuracy (77%), Se (72%), and PPV (82%). However, the best algorithm for this model was XGB for Whites and naïve Bayes for minorities. Among Whites, the best models were the MRCI/APACHE II, SOFA, and MRCI/SOFA models, in terms of precision (82%), precision accuracy (83%), and classification accuracy (90%), respectively. Thus, the inclusion of APACHE II improved the Se, whereas the inclusion of SOFA improved the PPV. Across minorities, the SOFA model was the most balanced model with a precision accuracy of 91%, a Se of 100%, and a precision of 82%. The addition of APACHE or SOFA to the home medication model improved the PPV (95%: 0.75–0.84; 95% CI: 0.82–0.89).

The home medication model performed better for the need for MV outcomes than for LOS, although it showed better metrics among Whites than among non-Whites. Indeed, precision accuracy reached 79%, Se, 78%, and precision, 82%, among Whites (Table 3). Among minorities, accuracy was 74%, Se was 74%, and precision was 76%. The contrast between the two understudied racial groups could not be more apparent than when the best models were examined. Among Whites, the MRCI/APACHE II outperformed with a precision accuracy of 91%, a Se of 89%, and a precision of 94%. However, among non-Whites, the SOFA and MRCI/SOFA models were the best.

## 4. Discussion

### 4.1. Key Findings

In the present study, we hypothesized that home medications could serve as predictors of health outcomes, either as a single factor (MRCI) or in combination with traditional scoring systems (APACHE II or SOFA). We also investigated the differential in predictive capabilities across races for these predictors. Our findings show that the home medication model predicts on average accurately ~70% of the time all three outcomes: ICU mortality, LOS, and need for MV. Along with SOFA, the MRCI model outperforms in predicting all three outcomes. In other words, without knowing the subsequent status of an ICU admitted patient, by just using the calculated MRCI (home medications) at admission one can predict with 70% accuracy mortality, LOS, and the need for MV. The combination of SOFA and MRCI vastly improved predictive accuracy among the general population. Moreover, another major finding of our study is that the MRCI was a better predictor of all three outcomes among Whites than it was among non-Whites. Among Whites, the predictive clinical accuracy of home MRCI reached 80%, while remaining at 70% for non-Whites for all outcomes. These findings provide justification to include home medication histories in the list of existing patient equity scoring systems such as APACHE II, SOFA, etc.

The SHAP values showed that low values of MRCI were associated with reduced mortality and LOS but an increased need for MV. Past research has corroborated our findings [41]. Herein, higher values of MRCI increased the odds of hospital mortality, as those with MRCI values above 14 were at least 1.84 times more likely to die during their hospital stay compared to those with MRCI values less than 5 [41]. A meta-analysis also found that high MRCI scores were associated with increased hospitalization hazards (hazard ratio = 1.20; 95% CI = 1.14 to 1.27) [42]. Across eight residential age care facilities in Australia, MRCI scores were shown to be positively correlated with LOS [18].

### 4.2. MRCI at Admission in Different Races

Interestingly, MRCI was a better predictor among Whites than it was among non-Whites. This could be due to a host of reasons, such as medication adherence and inaccurate medication history record, source of medication history record, class/covariate imbalance, and low sample size. There is a dearth of data on the variation of MRCI among racial groups, and whenever available, this evidence appears to be stratified by disease or confined to medication use. For example, past research in systemic lupus erythematosus showed that Blacks typically have higher medication regimen complexity scores compared to Whites [24]. Among Blacks, high MRCI scores are correlated with non-adherence to drug regimens [43]. This non-adherence may be due to a greater propensity for “pharmacy deserts” in low-income communities or medically underserved areas where minorities typically reside [44]. Geographical access to pharmacists’ services to manage regimens is thus not readily available for minorities [44,45]. As adherence/non-adherence may reflect MRCI scores [26,46], it may explain why Whites adhere more to drug regimens, and why this index is more reliable in predicting inpatient outcomes in this subpopulation.

### 4.3. Role of Critical Care Pharmacist

In the acute care setting, clinical pharmacists are considered medication experts and have extensive training in pharmacotherapeutics to provide comprehensive medication management to both patients and members of the interdisciplinary care team. It has been well documented that the addition of pharmacists providing direct patient care has led to a reduction in preventable adverse drug events, fewer transfers to the intensive care unit (ICU), and a reduction in length of stay (LOS) [47,48]. Additionally, the incorporation of critical care pharmacists within the interdisciplinary care team has led to improved patient outcomes, including mortality, ICU length of stay in mixed ICUs, and preventable/nonpreventable adverse drug events [49]. Clinical pharmacists play an essential role in evaluating home medication regimens for complexity in terms of appropriateness of use, safety of continuation during hospitalization, and frequent regimen modifications (i.e., dose reduction) during transitions of care. A 2017 study highlighted the importance of pharmacist intervention and communication during transitions of care from hospitalization to the community by demonstrating a 36% reduction in medication-related hospitalizations among the elderly population [50]. Therefore, the use of an MRCI scoring system by clinical pharmacists remains essential to improving the quality of pharmaceutical care delivered.

### 4.4. Different Biases in Predicting Clinical Outcomes

Racial bias, often encountered in data-driven algorithms for healthcare, remains a known challenge to delivering equitable, high-quality healthcare. Label choice-induced bias has been widely documented and has been well described noting the discrepancy between unobserved optimal prediction and a prediction trained on an observed label [51]. This type of bias typically springs from mismeasurement and human judgment/interpretation [52]. Data obtained from electronic health records (EHR) or claims databases reflect unaccounted clinical errors, which can lead to mismeasurement and bias in predictions. In this study, the measurements of MRCI were obtained using EHR data. However, a more informative score would be a version of the MRCI calculated using pharmacy insurance claims. A recent study examined the correlation between an EHR-based MRCI and a pharmacy claims-based medication complexity tool [53]. The authors claimed that the claims-based tool would better capture all pharmacy encounters compared to the EHR, which reflected mostly provider interactions. The association between the EHR and claims data was significantly higher among a subset of patients with similar counts of records between the EHR and claims. The association was lower for patients with large discrepancies between medication orders captured in the EHR and pharmacy claims. It is thus possible that patients with better access to care may have more reliable counts of EHR records, which when considered may closely reflect MRCI scores at home, than patients who mostly utilize medical services sporadically and mostly through emergency services (minorities) [54].

Other sources of racial bias include demographics, comorbidity imbalances in the dataset, and other societal and systemic sources [55]. A significant statistical disproportion was found with Blacks in our study making up 35.4% of the sample. Research has also shown that minorities are often, compared to Whites, at increased risk for chronic diseases [56]. Societal and systemic sources describe the unequal probability of having members of a specific racial group as a patient in our sample, perhaps due to strict patient selection or assessment of patient compliance, baseline health status, and comorbidities [55]. The mistrust of minorities with respect to medical providers could also play a crucial role here [57].

In the present study, the SOFA model was the best for Whites (mortality outcome), whereas the MRCI/APACHE II model had a better predictive ability for LOS and the need for MV outcomes. Among non-Whites, the SOFA and MRCI/SOFA models were the best models across all outcomes. Thus, for Whites, the addition of APACHE II to the base model seemed to enhance the predictive ability of the algorithm, whereas for non-Whites, the same effect was observed following the inclusion of SOFA scores. APACHE II is a disease severity classification system that focuses on the severity of disease based on physiological values. SOFA is an organ failure score based on the degree of dysfunction of six critical organs, and it is directly related to inpatient outcomes in critically ill patients. We hypothesize that SOFA is a better predictor among minorities because of the latter’s propensity to use emergency rooms as a primary resort for medical services [58], as a result of critical/worse (than Whites) health status. This is then better reflected across SOFA than APACHE II scores.

### 4.5. Implications of the Study

The present study highlights the benefits and advantages of using home medication histories for determining prognosis among a critically ill population. Identifying high-risk patients using this tool could substantially reduce the demand for critical care pharmacists, lower the overall costs of medical procedures, and also improve patient outcomes. As study strengths, this work helped set MRCI as a single or composite-value predictor for mortality, LOS, and the need for MV among hospitalized patients. Our results highlight the importance of critically evaluating home medication use to accurately and swiftly depict the prognosis to assist family members in making the challenging decision(s) to escalate care for their loved ones. Despite the widespread use of MRCI scores, researchers have yet to incorporate home medication use in a predictive model to forecast relevant outcomes. In terms of real-world implications, the bedside clinician is encouraged to evaluate the patient specific MRCI profile prior to making medical decisions. This in turn will positively impact patient safety and mitigate unnecessary risks. Incorporation of the MRCI has the potential to reinforce the current, yet not widely adopted, effort toward improved medication reconciliation, particularly upon transitions of care where most medication errors occur. Further, for critical care pharmacists, this tool could also serve to clearly identify the patients at the highest risk of treatment failure (patient readmission), in need of intense follow-up post-ICU admission, and equitably distribute future resources necessary to improve post-ICU and long-term care [59,60].

This study also promotes the use of explainable machine learning to investigate ICU outcomes, as informed by past research [61]. Further, it also emphasized the utility of the SMOTE technique in helping overcome class imbalance and producing robust results. It also highlighted the proficiency of several algorithms, specifically XGB, as well-suited to predict inpatient outcomes. XGB is a scalable and accurate implementation of gradient-boosting machines specifically designed to push the limits of boosted tree algorithms. XGB often produces the best predictive performances and processing times across several algorithms, and it has also been shown to work well in small samples.

### 4.6. Limitations

A few notable limitations are worth noting. First, we acknowledge that MRCI may be described as a surrogate index for disease severity. Second, racial bias on the basis of statistical disproportions limited the use of non-augmented (SMOTE) data in our algorithm. Despite the prowess of SMOTE, one of its major drawbacks is overfitting through the random synthesis of minority data while taking little to no account of the significance of the majority class [62]. Further, we were limited in the use of other covariates of importance in the dataset. Although MRCI alone could predict 70% of the time, typically the best predictors record accuracies of 90%. Perhaps a larger data set would yield better results. This study’s findings are also limited in their generalizability, as the data were obtained from a single medical center and may not be representative of the general population.

## 5. Conclusions

Home medication history is a robust predictor of ICU mortality, LOS, and the need for MV. The predictive capabilities of traditional patient acuity scoring systems improve with the addition of MRCI scores at admission. Racial attributes appear to determine the degree of importance that medication regimen complexity occupies with respect to clinical outcomes. In spite of our encouraging findings, future research should aim to validate these findings not only across larger samples but across subpopulations of varying demographic characteristics. Indeed, incorporating strategies to mitigate racial bias and introduce fairness in predictive algorithms could help pharmacists better attend to patients and, in turn, improve outcomes and reduce overall healthcare costs.

## Figures and Tables

**Figure 1 ijerph-20-03760-f001:**
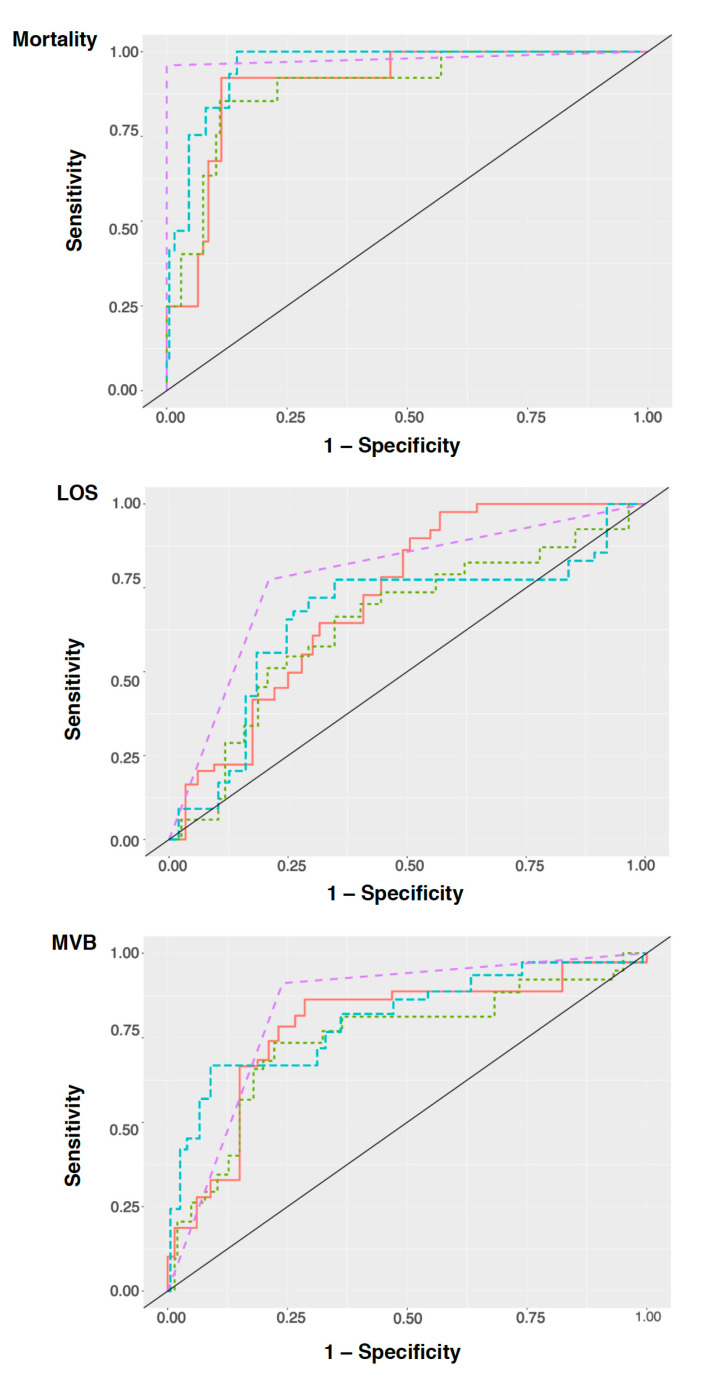
Area under the curve for mortality (**top**), LOS (**middle**); and MV (**bottom**); Algorithm colors–Purple: XGBoost, red: logistic classifier, Green: naïve Bayes, blue: random forest.; LOS: length of stay, MV: need for mechanical ventilation.

**Table 1 ijerph-20-03760-t001:** Demographic characteristics of low and high MRCI groups.

	Measure	Low MRCI	High MRCI	All	*p*-Value
Total	Total (%)	239 (74.9)	80 (25.1)	319	0.023 *
Demographics	Age, Median (IQR)	62 (50–76)	62.2 (53.8–70.2)	62 (51–75.8)	0.44
	BMI, Median (IQR)	28.8 (23–32)	29.3 (23–34)	28.9 (23–32)	0.21
	Male, *N* (%)	133 (56)	43 (54)	176 (55)	0.84
	White, *N* (%)	162 (68)	44 (55)	206 (65)	0.048 *
	Non-White, *N* (%)	77 (32)	36 (45)	113 (35)	0.041 *
	Hispanic, *N* (%)	20 (8)	17 (21)	37 (12)	0.004 **
Predisposing factors	Hypertension, *N* (%)	77 (32.2)	30 (37.5)	107 (33.5)	0.47
	Acute renal failure, *N* (%)	72 (30.1)	25 (31.3)	97 (30.4)	0.96
	Myocardial infarction, *N*(%)	54 (22.6)	24 (30)	79 (24.8)	0.24
	Metabolic encephalopathy, *N* (%)	41 (17.2)	16 (20)	59 (18.5)	0.68
	Long term insulin use	40 (16.7)	21 (26.3)	62 (19.4)	0.088
Traditional scoring systems	APACHE II, Median (IQR)	18.6 (13–22.8)	21.6 (16–26)	19.4 (14–23.8)	0.912
	GSC, Median (IQR)	12 (9–15)	11.1 (8–15)	11.7 (9–15)	0.39
	SAPS II, Median (IQR)	16 (6.2–24)	15.8 (9–21.5)	15.9 (7–23.8)	0.524
Outcomes	Mortality (%)	53 (22)	21 (26)	74 (23)	0.565
	LOS, Median (IQR)	113.2 (21–109)	106.5 (22–130.2)	111.5 (21–111.8)	0.084
	MV, Median (IQR)	65 (0–25.5)	47.9 (0–28)	60.7 (0–25.5)	0.179

IQR: Interquartile range; N: sample size within strata; GSC: Glasgow Coma Scale; LOS: length of stay; MV: mechanical ventilation; non-Whites encompasses Blacks, Asians, and all others. *: 0.01 < *p* < 0.05, **: 0.001 < *p* < 0.01.

**Table 2 ijerph-20-03760-t002:** Performance of machine learning algorithms across several models and outcomes.

Model and Outcome	AUC	Sensitivity	Specificity	PPV	NPV
Outcome: Mortality					
Home Medication Model (MRCI)	0.69 (0.66–0.73)	0.73 (0.68–0.78)	0.67 (0.63–0.72)	0.59 (0.54–0.64)	0.80 (0.75–0.84)
Admission Model (APACHE II))	0.91 (0.89–0.93)	0.88 (0.84–0.91)	0.94 (0.91–0.96)	0.94 (0.91–0.96)	0.87 (0.83–0.91)
MRCI and APACHE II Model	0.91 (0.89–0.93)	0.93 (0.90–0.96)	0.89 (0.85–0.92)	0.88 (0.84–0.91)	0.94 (0.91–0.96)
SOFA Model	0.97 (0.96–0.98)	0.94 (0.91–0.96)	1.00 (0.99–1.00)	1.00 (0.99–1.00)	0.94 (0.91–0.96)
MRCI and SOFA Model	0.98 (0.97–0.99)	0.96 (0.93–0.98)	1.00 (0.99–1.00)	1.00 (0.99–1.00)	0.96 (0.93–0.98)
Outcome: Length of stay					
Home Medication Model (MRCI)	0.68 (0.64–0.71)	0.64 (0.59–0.68)	0.72 (0.67–0.77)	0.76 (0.71–0.80)	0.60 (0.55–0.65)
Admission Model (APACHE II))	0.79 (0.76–0.82)	0.80 (0.75–0.84)	0.78 (0.73–0.82)	0.75 (0.70–0.79)	0.82 (0.78–0.86)
MRCI and APACHE II Model	0.77 (0.74–0.80)	0.68 (0.63–0.73)	0.70 (0.65–0.75)	0.68 (0.63–0.73)	0.70 (0.65–0.75)
SOFA Model	0.80 (0.77–0.83)	0.82 (0.77–0.86)	0.79 (0.74–0.83)	0.76 (0.71–0.80)	0.84 (0.80–0.88)
MRCI and SOFA Model	0.79 (0.76–0.82)	0.79 (0.74–0.83)	0.80 (0.76–0.84)	0.79 (0.74–0.83)	0.80 (0.75–0.84)
Outcome: Need for mechanical ventilation					
Home Medication Model (MRCI)	0.75 (0.72–0.78)	0.78 (0.73–0.82)	0.72 (0.67–0.77)	0.72 (0.67–0.77)	0.78 (0.73–0.82)
Admission Model (APACHE II))	0.77 (0.74–0.80)	0.75 (0.71–0.80)	0.79 (0.74–0.83)	0.78 (0.73–0.82)	0.76 (0.72–0.81)
MRCI and APACHE II Model	0.77 (0.73–0.80)	0.76 (0.72–0.80)	0.77 (0.72–0.82)	0.80 (0.76–0.84)	0.73 (0.68–0.77)
SOFA Model	0.85 (0.83–0.88)	0.88 (0.83–0.91)	0.84 (0.80–0.87)	0.81 (0.77–0.85)	0.89 (0.86–0.92)
MRCI and SOFA Model	0.87 (0.85–0.90)	0.90 (0.86–0.93)	0.86 (0.82–0.89)	0.84 (0.79–0.87)	0.91 (0.88–0.94)

Note: AUC: area under the receiver operating characteristic curve; PPV: positive predictive value; NPV: negative predictive value.

**Table 3 ijerph-20-03760-t003:** Performance of machine learning algorithms across several models and outcomes in a subpopulation of Whites.

Model and Outcome	AUC	Sensitivity	Specificity	PPV	NPV
Outcome: Mortality					
Home Medication Model (MRCI)	0.81 (0.78–0.84)	0.83 (0.79–0.87)	0.79 (0.75–0.83)	0.78 (0.74–0.82)	0.84 (0.79–0.87)
Admission Model (APACHE II) ^†^	0.80 (0.77–0.84)	0.86 (0.81–0.90)	0.73 (0.68–0.77)	0.66 (0.61–0.71)	0.89 (0.86–0.92)
MRCI and APACHE II Model	0.81 (0.78–0.84)	0.80 (0.76–0.84)	0.82 (0.78–0.86)	0.83 (0.79–0.87)	0.79 (0.74–0.83)
SOFA Model	0.76 (0.72–0.80)	0.67 (0.63–0.72)	0.88 (0.83–0.91)	0.90 (0.86–0.93)	0.62 (0.57–0.67)
MRCI and SOFA Model	0.88 (0.85–0.90)	0.86 (0.82–0.93)	0.90 (0.86–0.93)	0.91 (0.87–0.93)	0.85 (0.80–0.88)
Outcome: Length of stay					
Home Medication Model (MRCI) *	0.77 (0.73, 0.81)	0.72 (0.68, 0.77)	0.78 (0.73, 0.82)	0.82 (0.77, 0.85)	0.67 (0.62, 0.72)
Admission Model (APACHE II)	0.78 (0.75–0.81)	0.82 (0.77–0.86)	0.74 (0.70–0.79)	0.73 (0.68–0.77)	0.83 (0.78–0.87)
MRCI and APACHE II Model	0.82 (0.79–0.85)	0.82 (0.77–0.85)	0.83 (0.78–0.86)	0.84 (0.80–0.88)	0.80 (0.75–0.84)
SOFA Model	0.83 (0.81–0.86)	0.81 (0.77–0.85)	0.87 (0.82–0.90)	0.88 (0.85–0.91)	0.78 (0.74–0.83)
MRCI and SOFA Model	0.81 (0.78–0.84)	0.77 (0.73–0.81)	0.88 (0.83–0.91)	0.90 (0.87–0.93)	0.72 (0.67–0.77)
Outcome: Need for mechanical ventilation					
Home Medication Model (MRCI)	0.79 (0.76, 0.82)	0.78 (0.73, 0.82)	0.80 (0.75, 0.84)	0.82 (0.78, 0.86)	0.75 (0.70, 0.80)
Admission Model (APACHE II)	0.88 (0.86–0.90)	0.94 (0.91–0.96)	0.83 (0.79–0.86)	0.82 (0.77–0.85)	0.94 (0.91–0.96)
MRCI and APACHE II Model	0.91 (0.89–0.93)	0.89 (0.86–0.92)	0.94 (0.90–0.96)	0.94 (0.91–0.96)	0.88 (0.84–0.91)
SOFA Model	0.72 (0.69–0.76)	0.71 (0.66–0.76)	0.74 (0.69–0.79)	0.77 (0.73–0.81)	0.67 (0.62–0.72)
MRCI and SOFA Model	0.80 (0.77–0.83)	0.76 (0.72–0.80)	0.84 (0.80–0.88)	0.87 (0.83–0.90)	0.72 (0.67–0.77)

Note: Results presented for the best algorithm (XGB) except with the following symbols: *: logistic classifier; ^†^: naïve Bayes. AUC: Area under the receiver operating characteristic curve; PPV: positive predictive value; NPV: negative predictive value.

**Table 4 ijerph-20-03760-t004:** Performance of machine learning algorithms across several models and outcomes in a subpopulation of non-Whites.

Model and Outcome	AUC	Sensitivity	Specificity	PPV	NPV
Outcome: Mortality					
Home Medication Model (MRCI)	0.73 (0.70–0.75)	1.00 (0.98–1.00)	0.61 (0.57–0.65)	0.45 (0.40–0.50)	1.00 (0.99–1.00)
Admission Model (APACHE II) *	0.80 (0.77–0.83)	0.70 (0.65–0.75)	0.83 (0.78–0.87)	0.84 (0.79–0.88)	0.69 (0.64–0.73)
MRCI and APACHE II Model *	0.78 (0.75–0.82)	0.76 (0.72–0.81)	0.85 (0.80–0.88)	0.84 (0.79–0.88)	0.77 (0.73–0.81)
SOFA Model *	0.84 (0.80–0.87)	0.76 (0.71–0.80)	0.90 (0.86–0.93)	0.90 (0.86–0.93)	0.75 (0.70–0.79)
MRCI and SOFA Model *	0.85 (0.81–0.88)	0.67 (0.63–0.72)	0.88 (0.83–0.91)	0.90 (0.86–0.93)	0.62 (0.57–0.67)
Outcome: Length of stay					
Home Medication Model (MRCI) ^†^	0.60 (0.55, 0.64)	0.59 (0.54, 0.63)	0.60 (0.54, 0.66)	0.69 (0.64, 0.73)	0.50 (0.44, 0.55)
Admission Model (APACHE II)	0.82 (0.80–0.85)	0.73 (0.69–0.77)	1.00 (0.98–1.00)	0.83 (0.78–0.87)	0.64 (0.59–0.69)
MRCI and APACHE II Model	0.76 (0.73–0.79)	0.70 (0.66–0.75)	0.83 (0.78–0.87)	0.80 (0.75–0.84)	0.64 (0.59–0.69)
SOFA Model	0.91 (0.89–0.93)	1.00 (0.99–1.00)	0.84 (0.80–0.87)	0.82 (0.77–0.86)	1.00 (0.99–1.00)
MRCI and SOFA Model	0.86 (0.83–0.88)	0.86 (0.82–0.89)	0.85 (0.81–0.89)	0.86 (0.82–0.89)	0.85 (0.81–0.89)
Outcome: Need for mechanical ventilation					
Home Medication Model (MRCI)	0.74 (0.71, 0.77)	0.74 (0.70, 0.79)	0.74 (0.69, 0.78)	0.76 (0.71, 0.80)	0.72 (0.67, 0.77)
Admission Model (APACHE II)	0.72 (0.68–0.75)	0.78 (0.72–0.82)	0.68 (0.63–0.72)	0.60 (0.54–0.65)	0.83 (0.79–0.87)
MRCI and APACHE II Model	0.76 (0.74–0.79)	0.67 (0.63–0.71)	1.00 (0.98–1.00)	1.00 (0.99–1.00)	0.52 (0.47–0.58)
SOFA Model	0.96 (0.94–0.97)	0.92 (0.89–0.92)	1.00 (0.99–1.00)	1.00 (0.99–1.00)	0.91 (0.88–0.94)
MRCI and SOFA Model	0.93 (0.91–0.95)	0.89 (0.85–0.92)	1.00 (0.99–1.00)	1.00 (0.99–1.00)	0.86 (0.82–0.90)

Note: Results presented for the best algorithm (XGB) except with the following symbols: *: logistic classifier; ^†^: naïve Bayes. AUC: area under the receiver operating characteristic curve; PPV: positive predictive value; NPV: negative predictive value.

## Data Availability

The data supporting the findings from the study are available from the corresponding author upon a reasonable request based on the approval by the medical institution.

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
