# Peer review of "Medication Regimen Complexity Index Score at Admission as a Predictor of Inpatient Outcomes: A Machine Learning Approach"

_ijerph, 2023, doi:10.3390/ijerph20043760_

Round 1
Reviewer 1 Report
I read your article with interest. The authors examine the development of the ICU outcomes prediction model using various machine learning methods. It is suitable for this topic of the article. The authors developed models applying standard methods and evaluated and compared each model. In the clinical setting, the prediction model of patient outcomes is helpful for medical staff.
This manuscript and study are interesting; however, I found some recommended points to be revised to improve this manuscript. I hope that the authors will consider my recommendation and modify it.
Major comments
My great concern is the issue of data used to predict model development. I suggest that the authors should make improve the data quality and quantity for machine learning. The reason is that you cannot obtain a reliable prediction model from data which is not good even if you apply such great machine learning methods.
1. Sample size
As the authors wrote in the limitation part, your data has a small sample size. Because of this problem, the analysis of race is limited to only White and Non-White. To achieve this analysis, the authors should collect enough cases. How many cases did the authors plan to? Have they calculated the sample size?
To improve this article more valuable, the authors should extend the study duration or consider collaborating with other institutes and adding more patients.
2. Collection of patients data
How did the authors prepare the patients' data? I could not understand enough from this manuscript. For instance,
• Procedures of extracting data from EHR; Who did it and how to do it?
• The existence of missing data; If it existed, how to deal with the data?
• The procedure of calculating MRCI; How was the information collected? Who calculates MRCI? Did several people double-check MRCI scores?
The reason why this information is required is that it has a great concern to occur some bias. Using the biased data, the developed prediction model will not be appropriate. (As you know, "Garbage in, garbage out.”)
3. Patient's Disease information
I could not find a kind of patient's disease information. I assume that the relationship between a disease treated by the drug at home (before entrance ICU) and a disease triggered to entrance ICU is essential information to consider patient's prognostic. If a patient enters ICU because of a worsening primary disease, this patient's MRCI is deemed to be high. And, if a patient enters ICU because of an unrelated primary disease, MRCI and outcome are assumed to have less relation. In other words, MRCI is just a surrogate index of the severity of the patient's primary disease in some patients.
On the other hand, If MRCI represents the difficulty of the patient's treatment, including their adherence, MRCI will relate to their prognosis of the event. However, I cannot judge based on this manuscript; therefore, I request detailed analyses and descriptions of the patient's primary disease, triggered disease for entering ICU, and relationship of MRCI. As the authors said, the ability to generalization in the limitation section and patients' disease is also related to the issue. I recommend adding hospital information.
4. Aim of developing a prediction model
Once you develop a prediction model, an application is also critical. What is the aim of this prediction model usage? I ask the authors to write about this point in more detail because I could not find it in this manuscript.
What is the advantage of this prediction model? I assume that calculating MRCI and predicting the outcome by using the model are needed a lot of costs. Thus the model should have merit over the expenses. And the variables used for the prediction model cannot modify when a patient enters ICU. I would like to know the detail of the association between an application of the model and clinical pharmacists.
I recommend applying the model to patients who are entered into ICU after this study period. It will be clarified that this model is not overfitting because these models have a high predictive performance.
5. Consideration of race factor
The relationship between outcome and race is interesting; however, I recommend to remove from this article. The reason is that this manuscript has not enough evidence to discuss this topic. As the authors wrote in the manuscript, they could compare only White and Non-White because of a small sample size. And also, data and analysis on other factors that might relate (financial, education, and so on) to the race problem are lacking. For these reasons, the discussion of race in the manuscript seems incomplete, and I am afraid that it will give a half-finished impression. I suggest the authors conduct further study, gather enough patient characteristic data, and analyze. This manuscript should concentrate on only developing a prediction model using MRCI.
Minor comments
1. Description of APACHE and SOFA
I recommend modifying the description of APACHE and SOFA in the Introduction section. Curiously, these scores are described in the same context as MRCI because APACHE and SOFA were developed to measure physiological specificity and judge severity, not predict outcomes. I agree with the concept of authors that are using APACHE or SOFA, and MRCI will improve prediction performance. Therefore, I suggest changing the explanation that a combination of APACHE or SOFA and MRCI is helpful in developing predicted outcomes.
2. Line 29- 35.
Please remove this paragraph.
3. Line 47- 49.
This data (reference 13) is outdated. Could you find current information?
I would like to know more about medication errors specifically and precisely.
4. Line 57- 59.
I recommend adding an explanation about MRCI (Why MRCI is the gold standard?). The "2.3.1. MRCI score: Key independent variable" description should be moved to the Introduction part. The authors should write details about collecting data for MRCI in the Methods part.
5. Line 134- 143 and Table S1
Table S1 shows models and input variables. It is helpful for understanding, but I suggest some improvements to this table.
・ Transpose rows and columns.
・ Demographic variables should not be summarized as "Demographics"; show all variables in rows.
・ Indicate input variables for models using a mark (ex., Fill checkmarks into cells used for modeling.)
6. Table 2
Some value seems different from Table 2 and the Supplemental file.
7. Line 501-.
The publisher of each article is written in reference. Is it following submission rules?
Thank you for giving me a great opportunity!
Author Response
"Please see the attachment."

Reviewer 2 Report
Very good work, I would like to thank the authors for this interesting contribution. The article is well-written, and the methodology is largely clear. It is also good to see that the authors are aware of possible limitations. I only have a few points to consider, please.
(1)
In the manuscript, please elaborate further on how SMOTE was used for balancing the dataset. It is not clear if SMOTE was applied before or after splitting the dataset into train/test sets.
(2)
The introduction or discussion should refer to studies that applied explainable ML techniques in the context of predicting ICU mortality. For example:
https://doi.org/10.1109/ICHI48887.2020.9374393
https://doi.org/10.3390/s21217125
(3)
The quality of figures should be improved, please.
Author Response
"Please see the attachment."

Reviewer 3 Report
This study examines the predictive value of MRCI score alone and in addition to "traditional" ICU patient scoring systems. There is value in better understanding how these scoring systems can be enhanced, and given the typical medication use of many patients who are admitted to the ICU, this is a logical addition to the use of existing tools.
Abstract
1) Line 22: The SHAP acronym has not been defined before it is used here.
Overall
2) Lines 29-35: Delete as this is part of the template.
Introduction
3) Lines 43-45: I'm not sure I understand what is meant by "clinical significance of medical history errors" - does this refer to errors that lead to harm or injury? Additionally, do these two sentences refer to different things? The first sentence discusses "medical history" while the second discusses "medication errors".
4) Lines 46-49: Reconcile the two different costs of errors presented here.
5) Line 54: Delete the second "ADE".
6) Line 69: Change "unexplored" to "explored".
7) Line 78: Add the word "potential" before "outcome of hospital stay".
Materials and Methods
8) Line 118-119: Not sure the description of mortality is needed, typically there is no other way to categorize mortality than as a binary variable.
9) Line 125: Change "Charlson score" to "Charlson comorbidity index score".
10) Lines 125-126: Delete "(White and Non-Whites)" as this is described in more detail later.
11) Line 127: Delete" (henceforth referred as MRCI)".
Results
12) Line 219: Delete the second "in".
13) Line 286: Edit the second half of the sentence to read: "patients were in need of MV."
Discussion
14) Line 342: Delete the first "the".
15) Line 373: Delete "as".
16) Line 391: Edit "MRC" to "MRCI".
17) Lines 402-404: A more accurate description may be that there are limitations with both EHR and pharmacy claims data. Pharmacy claims data is also limited as it will not include medications paid for out-of-pocket.
18) Line 438: Delete "shows".
19) Line 439: Add "a" before "critically ill population".
20) Lines 440-441: It is unclear how identifying high-risk patients will reduce the demand for critical care pharmacists (won't it increase the demand as pharmacists can assist in addressing the needs of these high-risk patients?) and lower the overall costs of medical procedures.
21) Line 441: Change "but" to "and".
Conclusions
22) Lines 464-465: Reverse the focus of this sentence to mention the predictive capabilities of the existing scoring systems first being further enhanced by adding MRCI.
23) Line 464: Change "or" to "of".
Author Response
"Please see the attachment."

Round 2
Reviewer 1 Report
Please confirm the attachment file.

Author Response
"Please see the attachment."
